# An ATP-Binding Cassette Transporter, LaABCB11, Contributes to Alkaloid Transport in *Lycoris aurea*

**DOI:** 10.3390/ijms222111458

**Published:** 2021-10-24

**Authors:** Rong Wang, Yantong Liu, Sheng Xu, Jie Li, Jiayu Zhou, Ren Wang

**Affiliations:** 1Institute of Botany, Jiangsu Province and Chinese Academy of Sciences, Nanjing 210014, China; njau2010216002@163.com (R.W.); liuyantong95@icloud.com (Y.L.); xusheng@cnbg.net (S.X.); LJ734820531@163.com (J.L.); zhoujiayu@cnbg.net (J.Z.); 2The Jiangsu Provincial Platform for Conservation and Utilization of Agricultural Germplasm, Nanjing 210014, China

**Keywords:** ABC transporter, lycorine, galantamine, *Lycoris aurea*

## Abstract

As a kind of Amaryllidaceae alkaloid which is accumulated in the species of Lycoris plants, lycorine has a range of physiological effects. The biosynthesis pathway of lycorine has been partly revealed, but the transport and accumulation mechanisms of lycorine have rarely been studied. In this study, an ATP-binding cassette (ABC) transporter from *Lycoris aurea* (L’Hér) Herb., namely LaABCB11, was cloned and functionally characterized. Heterologous expression showed that LaABCB11 transported lycorine in an outward direction, increased the tolerance of yeast cells to lycorine, and caused a lower lycorine accumulation in transformants than control or mutant in yeast. LaABCB11 is associated with the plasma membrane, and in situ hybridization indicated that *LaABCB11* was mainly expressed in the phloem of leaves and bulbs, as well as in the cortical cells of roots. These findings suggest that LaABCB11 functions as a lycorine transport and it might be related to the translocation and accumulation of lycorine from the leaves and bulbs to the roots.

## 1. Introduction

Plants of the Amaryllidaceae family produce particular types of alkaloids, including lycorine, haemanthamine, galantamine and other Amaryllidaceae alkaloids; these alkaloids are generally used for defense purposes against pathogens, insects, and herbivores. Some Amaryllidaceae alkaloids are also beneficial for humans, having pharmacological uses such as the anti-cancer [1], anti-bacterial activity, and antiviral impact against coronavirus and Poliomyelitis virus [2] of lycorine. Galantamine, an Amaryllidaceae alkaloid, has been clinically used to treat the symptoms of Alzheimer’s disease [3]. The medical and economic values of Lycoris plants are poorly investigated, and information regarding the genes involved in the whole biosynthesis pathway remains fragmentary [4,5]. Only several enzymes involved in the biosynthetic pathway of Amaryllidaceae alkaloids have been identified and characterized to date. The core biosynthesis pathway of Amaryllidaceae alkaloids starts with tyramine, which is decarboxylated from tyrosine by decarboxylase (TYDC) [6], and 3, 4-dihyroxybenzaldehyde, which derives from phenylalanine by ammonia-lyase (PAL) [3]. Then, norbelladine, which is the base carbon scaffold of Amaryllidaceae alkaloids, is condensed from tyramine and 3, 4-dihyroxybenzaldehyde by enzyme norbelladine synthase [7]. Following the condensation reaction, norbelladine is methylated by norbelladine 4′-*O*-methyltransferase (N4OMT) to 4′-*O*-methylnorbelladine [4], which is the scaffold incorporating several types of Amaryllidaceae alkaloids, such as galantamine, lycorine and crinine [8]. After that, 4-*O*-methylnorbelladine is coupled, in a *phenol-phenol*’ manner, to generate the crucial intermediate structure. As the position of *C-C* bond can be in three different ways (i.e., *para-ortho*’, *ortho-para*’ or *para-para*’), the enzyme-catalyzed *C-C* phenol coupling is the decisive factor in generating the crucial core skeleton [5]. However, downstream of 4′-*O*-methylnorbelladine, only one gene has been reported so far: cytochrome P450 monooxygenase 96T1 (CYP96T1) of *Narcssus* sp. *aff. pseudonarcissus*. NpCYP96T1 has been shown to be capable of forming the *para-para*’ *C-C* phenol coupling production, noroxomaritidine. In the meantime, NpCYP96T1 has also been found to catalyze the formation of *para-ortho*’ production, *N*-demethylnarweding, even at a very low level [9]. Of these two intermediate products, noroxomaritidine is further modified into the metabolite pathway of crinine and narciclasine, and *N*-demethylnarweding is modified into the lycorine-type Amaryllidaceae alkaloid biosynthesis pathway [10].

Generally, plant secondary metabolites—especially alkaloids and their unique catalyzed enzymes—are spatially separated [11]; apparently, the tight and crucial transport mechanism provides important machinery for alkaloid biosynthesis and accumulation. To date, several secondary metabolites and important intermediate transporters have been found in plants [11,12]. Transporter families, including ATP-binding cassette (ABC), multidrug and toxin extrusion (MATE), purine uptake permease-like (PUP), and so on, have been demonstrated to be involved in the inter-organ, inter-cellular, and intra-cellular transport of such compounds in plants [13,14,15]. Three transporters of berberine in *Coptis japonica* (Thunb.) Makino have been well-documented. Among them, CjABCB1 (CjMDR1) and CjABCB2 are responsible for mediating berberine transport from the synthesis site (root) to the stock site (rhizome) [16]. Besides, CjMATE1 has been demonstrated to functions as a vacuolar transporter of berberine [17]. In another medicinal plant, *Catharanthus roseus* (L) G. Don, it has been reported that a plasma-membrane localized ABCG transporter CrTPT2 takes part in the efflux of catharanthine to the leaf surface, and the expression level of *CrTPT2* directly affects the yield of catharanthine-vindoline dimers, vinblastine [11]. Recently, in the biotechnology of synthesizing plant natural products, the strategy of addressing and introducing transporters into engineered microbes has been shown to be effective in approving product yield. Srinivasan and Smolke (2021) have identified two transporters—PUP1 and lactose permease-like transporter 1 (LP1)—in *Atropa belladonna* L., a tropane alkaloid-producing plant. Both of these transporters, along with two other transporters (NtJAT1 and NtMATE2), have been introduced into engineered yeast to achieve an over 100-fold yield of hyoscyamine and scopolamine production [18].

In this study, by searching the transcriptome data of *L. aurea*, we identified an ABC transporter, LaABCB11, and analyzed its function using yeast mutant cells. In addition, we also analyzed the tissue-specific localization of *LaABCB11* in order to characterize the physiological function of this transporter.

## 2. Results

### 2.1. Isolation of Lycorine Transporter from L. aurea and Lycorine Content in Different Tissues

The induction of lycorine under methyl jasmonate (MeJA) treatment suggested that lycorine transporters might be up-regulated by MeJA. By inspection of the transcriptome profiling of *L. aurea* seedlings with MeJA treatment [6], several contig coding transporter candidates have been identified, including a typical ABC transporter, which is annotated *LaABCB11*. It covers an open reading frame representing 1290 amino acids (the sequence is available in Appendix A), containing two transmembrane domains (TMDs) and two nucleotide-binding domains (NBD). The protein sequence of LaABCB11 was phylogenetically analyzed with other ABC transporter homologous from other species, using the MEGA 7 software. The phylogenetic tree showed that LaABCB11 shares high identity to other known plant ABCBs, and is located close to ABCB11 from *Dendrobium catenatum* Lindley and *Elaeis guineensis* Jacq. (Figure 1A). We performed Real-time PCR to determine *LaABCB11* expression in different tissues and under MeJA treatment. The results showed that, at leaf stage, *LaABCB11* was mainly expressed in roots and leaves, rather than in bulbs (Figure 1B). We also investigated the expression of *LaABCB11* under MeJA treatment; as shown in Figure 1C, in leaves, *LaABCB11* transcription was increased approximately 2- to 7-fold from 0 to 36 h under 100 mM MeJA treatment. Meanwhile, the lycorine content in different tissues of *L. aurea* seedlings were measured. The result showed that the highest content of lycorine was found in roots, whereas the lowest content was found in bulbs (Figure 1D).

### 2.2. LaABCB11 Functions as a Lycorine Efflux in Yeast

The functional diversity of ABC transporters has been investigated in various heterologous expression systems, such as in yeast, animal, or plant cells [19]. In order to examine whether LaABCB11 function as a lycorine transporter, *LaABCB11* was constitutively expressed under the PMA promotor [11] of the pDR196 vector in the yeast mutant strain AD12345678, which is lacking eight yeast ABC transporter genes [20]. In the presence of lycorine, the LaABCB11-transformed yeast cells (pDR196-LaABCB11) showed an obvious tolerance to lycorine, compared with the vector control (pDR196) (Figure 2A). This suggests that lycorine was recognized as a substrate of LaABCB11 and resulted in the growth inhibition of LaABCB11-transformed yeast cells. Then, to characterize LaABCB11 activity, the lycorine content in yeast was quantitatively analyzed. We used lipid SD medium (- uracil) containing 0.3 mM lycorine to treat the yeast cells. The results showed that, after 2–12 h lycorine treatment, the lycorine contents of each transformant were significantly different, with the yeast expressing LaABCB11 (pDR-LaABCB11) always accumulating a higher content than the empty control (pDR196) and the mutated version of LaABCB11 (pDR196-ΔLaABCB11), in which both ATP-binding motifs were deleted [21] (Figure 2B). These results indicated that LaABCB11 functions as a lycorine efflux transporter, and that the transport activity is ATP-dependent.

### 2.3. Sub-Cellular Localization of LaABCB11

To further study the function of LaABCB11, we assessed the sub-cellular localization of LaABCB11-GFP in both *Arabidopsis thaliana* (L.) Heynh and *Nicotiana*
*benthamiana* Domin. First, the fusion protein LaABCB11-GFP was expressed in transgenic *A. thaliana* lines, with most of the GFP signal showing at the plasma membrane of root cells (Figure 3A). Then, LaABCB11-GFP with the plasma membrane marker CBL-mCherry [22] were transiently co-expressed in *N. benthamiana* leaves. Most of the GFP signal overlapped with mCherry (Figure 3B). Only a small fluorescent signal was also observed inside the cells (Figure 3B), which may reflect endoplasmic reticulum localization, where the protein was assembled [23]. These results suggested that the LaABCB11-GFP fusion protein was associated with the plasma membrane.

### 2.4. LaABCB11 Expression Pattern in Accordance with Its Function

The expression pattern of *LaABCB11* was analyzed using the in situ hybridization technique in 3-mouth-old leaf stage *L. aurea* seedling grown in an illuminating incubator. Paraffin-embedded cross-sections of roots, bulbs and leaves were hybridized with specific anti-sense and sense RNA probes of *LaABCB11*. In leaves, labeling of *LaABCB11* expression with the anti-sense probe was detected in stele, mainly in phloem (Figure 4A). In bulbs, the probe signal was also detected in phloem. However, when compared with leaves, the staining was lighter in bulbs, suggesting that the expression level of *LaABCB11* in bulbs was lower than that in leaves (Figure 4B). The labeling signal of the probe was mainly detected in cortex cells of roots, and there was no staining in the stele (Figure 4C). Meanwhile, the negative controls using *LaABCB11* sense probes did not label any significant signal (Figure 4D–F). In summary, *LaABCB11* was mainly expressed in the stele tissue of leaves and bulbs, and in the cortex of roots.

### 2.5. Substrate Specificity of LaABCB11

We also examined other possible substrates of LaABCB11 with other Amaryllidaceae alkaloids, including galantamine, narciclasine and tazettine. Our results showed that the LaABCB11 transformant appeared to accumulate less galantamine than vector control (Figure 5A). On the other hand, there was no difference in the uptake of narciclasine or tazettine between the control and LaABCB11 transformant (Figure 5B,C). These data suggest that galantamine could also be recognized by LaABCB11 as a substrate and exported out of the yeast cells; however, neither narciclasine nor tazettine is not recognized, indicating that LaABCB11 had substrate specificity to lycorine and galantamine.

## 3. Discussion

As many plant nature products have interesting bioactivity, these compounds have served as an important source of natural medications for humans. In fact, in Chinese traditional medicine, officinal plants have been used to treat various illnesses for thousands of years. With the development of biotechnology, the biosynthesis pathways of some important compounds have been fully elucidated [24], and some transporters belonging to different families have been identified and characterized. The B sub-family members of ABC transporters have been shown to be involved in auxin, berberine, and sesquiterpene pyridine transport [19,25]. In this article, we identified one ABC transporter from *L. aurea*, LaABCB11, which carries out the transmembrane transport of at least two kinds of Amaryllidaceae alkaloid (lycorine and galantamine); furthermore, it is also involved in the polar transport of these alkaloids between organs.

From earlier studies, some key genes in the lycorine and galantamine biosynthetic pathways have been found to be mainly expressed in flower and flower stalk tissue of mature *L. aurea*, whereas lycorine and galantamine are preferentially accumulated in the flower, bulb and root [26]. This phenomenon may imply that lycorine and galantamine are synthesized and accumulated in separate tissues. The secondary metabolites do not accumulate at the position producing, they may be transported to other tissues in a vascular manner, for protection, metabolism and degradation [27]. The application of MeJA clearly accelerated the accumulation of Amaryllidaceae alkaloids in *Lycoris chinensis* Traub seedlings [28]. Transporter expression is activated through the application of an exogenous substrate [11]. By searching the MeJA response transcriptome of *L. aurea* [6], we identified several candidate transporters including a typical ABC transporter member, *LaABCB11*. Transport assay using yeast cells have been wildly applied in the study of transporter [19]. In this study, the yeast transformant pDR196-LaABCB11 was more tolerant to lycorine in the drug sensitive assay and accumulated less alkaloid than the empty vector (Figure 2). As the nucleotide-binding domain (NBD) contains conserved motifs involved in Mg-ATP binding and hydrolysis [19], when both ATP-binding motifs were deleted, the mutant ΔLaABCB11 failed to export lycorine out of the yeast cell (Figure 2B). Meanwhile, we also determined that LaABCB11 localized primarily in the plasma membrane (Figure 3). All of these results suggest that LaABCB11 is ATP-dependent and functions as a lycorine efflux transporter.

The transporters expressed in vascular tissues seem to act in the polar transport of the substrate [29]. For example, CjMDR1 localized in the rhizome xylem has been shown to be involved in the translocation of berberine from root to rhizome [16]. *LaABCB11* was mainly expressed in the phloem of the leaves and bulbs (Figure 4A,B). These data suggest that LaABCB11 might takes part in the polar transport of lycorine through phloem tissue from leaves and bulbs to roots for accumulation (Figure 1D). In [30], it has been detected that galantamine is mainly stored in the apoplastic part, the cell wall. We also showed that *LaABCB11* is also expressed in root tissue, restricted to periphery cell (Figure 1B and Figure 4C). Plant roots may release active material into the apoplast and soil, in order to protect themselves against pathogenic infections [25]. We suspected that LaABCB11 takes part in the export of lycorine in root cells to the exoplasmic space, even soil, in order to avoid feedback inhibition of biosynthesis enzymes and help the plant against pathogenic infections [16].

The transporters typically show a broad substrate specificity, such as MDR-type ABC transporter P-glycoprotein [19]. The substrate specificity of LaABCB11 seemed not to be exclusively specific to lycorine, but also recognized galantamine (Figure 5A), at least. However, other Amaryllidaceae alkaloids—tazettine and narciclasine—were not recognized by this transporter (Figure 5B,C). Whether LaABCB11 can transport other kinds of substrates requires further investigation. Furthermore, the other transporter genes involved in lycorine transport should be found and characterized in the future works.

## 4. Materials and Methods

### 4.1. Plant Materials

*L. aurea* seeds were planted in an illuminating incubator with 16/8 h photoperiod at 25/18 °C for 3 months. The seedlings were treated with 100 µM MeJA in 0.02% DMSO for 0, 6, 12, 24 h, and 0.02% DMSO was used as vehicle control [26]. *A. thaliana* and *N. benthamiana* seeds were germinated and grown in sterilized soil with 16/8 h photoperiod at 25/20 °C.

### 4.2. Isolation of LaABCGB11 cDNA

Following the transcriptome database of *L. aurea*, the *LaABCB11* cDNA region was amplified using the relevant set of primers (*ABCB11*-s and *ABCB11*-a, Appendix A), and cloned into the T-easy vector (Invitrogen, Shanghai, China) for sequencing by Sangon Shanghai.

### 4.3. RNA Isolation, cDNA Synthesis and qRT-PCR

RNA was extracted using RNAiso Plus reagent (Takara, Dalian, China), following the manufacturer’s instructions. The synthesis of first-strand cDNA was performed as previously described [31], using the PrimerScript RT Reagent Kit with gDNA Eraser (Takara). The Primers RT-s and RT-a were used for qRT-PCR. The gene LaTIP4 [32] was used as the internal gene, with the primers TIP4-s and TIP4-a. qRT-PCR was performed using the SYBR Green master mix (Takara) on the Jena system (Jena, Germany), under the following conditions: 94 °C for 2 min; 35 cycles of 94 °C for 15 s, 60 °C for 20 s, and 72 °C for 30 s; and a final extension at 72 °C for 2 min.

### 4.4. Functional Analysis of LaABCB11 in Yeast

Five segments of ΔLaABCB11 separated by four ATP-binding motifs Walker A and Walker B [33] in NBD were obtained by PCR using five primer pairs listed in Appendix A. The full length of ΔLaABCB11 was cloned by overlap PCR using primers 11-1s/11-5a, with the five segments as template. The full lengths of LaABCB11 and ΔLaABCB11 were subcloned using primers 196-11-s/196-11-a and linked into the Spe I and EcoR I sites of the vector pDR196, using the ClonExpress Ultra One Step Cloning Kit (Vazyme, Nanjing, China) to construct the plasmid pDR196-LaABCB11 and pDR196-ΔLaABCB11. The constructions and the empty vector pDR196 were then transformed into yeast strain AD12345678 using the lithium acetate method [34]. The transformed yeast colonies were verified by PCR and sequencing, and the positive transformants were prepared for further assays.

The drug sensitivity of the yeast cell was tested by spotting the yeast transformant’s cell onto agar plates containing 100 µM lycorine or DMSO. The transformant cells were pre-cultured overnight in SD medium (-uracil). The cultures were diluted to the same density, OD_600_ = 0.5. Five microliters of cells were spotted onto the plates and incubated at 30 ℃ for 2 days.

Lycorine export assays were conducted following the method of [11]. The yeast transformants were cultured in SD medium (-uracil) until OD_600_ = 1.0. Then, the cells were collected and resuspended in 1/2 SD medium (-uracil) containing 300 µM lycorine (dissolved with DMSO) or DMSO, and harvested at indicated times. The yeast cells were washed twice with sterile ddH_2_O, disrupted with glass beads for 15 min at 30 Hz, and centrifuged. The supernatants were used for further HPLC analysis.

Different alkaloids (30 µM) were added to 4 mL of transformed yeast cells culture (OD_600_ = 1.0) for substrate specificity assays. After 3 h incubation with shaking (150 rpm) at 30 °C, the yeast cells were centrifuged, and the medium was collected for HPLC analysis. The amounts of substrates inside the yeast cells were calculated.

### 4.5. Transient Expression in N. benthamiana Leaves

For transient expression of LaABCB11-GFP fusions, the full-length CDs region of *LaABCB11* was cloned using primer 1301-11-s and 1301-11-a and linked into the BamH I and Xba I sites of the vector pCAMBIA1301-GFP. The construct was transformed into *Agrobacterium tumefaciens* strain EHA105. Four-week-old tobacco seedling leaves were infiltrated using the transformed *A. tumefaciens* cells, following the method described in [35], and were placed in a culture incubator under 16 h/8 h light/night cycles at 25 °C for 48–72 h. The infiltrated leaves were examined using exciter confocal laser-scanning microscopy (Zeiss LSM780) with an excitation wavelength of 488 nm and a 505–530 nm bandpass filter.

### 4.6. Transformation of A. thaliana

The construct pCAMBIA1301-11-GFP mentioned above was transformed into *A. thaliana* through *A. tumefaciens* EHA105 following the protocol reported by Bechtold and Pelletier [36]. The transformant lines were screened using 30 mg/L hygromycin.

### 4.7. In Situ Hybridization

The primer pairs probe-s/probe-T7-a and probe-T7-s/probe-a (Appendix A) were designed for PCR amplification of DNA template for sense and anti-sense probes of *LaABCB11*. The probes were generated and digoxigenin-labeled using the DIG RNA labeling kit (Roche, Mannheim, Germany). Tissue preparation was carried out following the method described by [37] with minor modification. Root, leaf and bulb samples were fixed with 4% paraformaldehyde overnight at 4 °C, then dehydrated using an ethanol series (ethanol 65%, 80%, 90%, 1 h, 100% twice for 1 h), and embedded with paraffin (Sigma, St. Louis, MO, USA) through several baths: xylene/paraffin (3:1), 8 h; xylene/paraffin (1:1), 8 h; xylene/paraffin (1:3), 8 h; paraffin, overnight. After that, the samples were cut into 10–15 µm thick sections and mounted on slides (Fisher, Pittsburgh, USA). Sections were dewaxed with xylene twice for 2 h, and rehydrated through an ethanol series (ethanol 100% twice for 1 h; 90%, 80%, 65%, 1 h). Slides were then treated with proteinase K (Invitrogen 0.1 unit/mL), washed with phosphate-buffered saline (PBS), and dehydrated again through an ethanol series. Then, the slides were hybridized with a digoxigenin-labeled RNA probe. Immunological detection of the hybridized probes was performed using a DIG Nucleic Acid Detection Kit (Roche), according to the manufacturer’s instruction. After color development, the slides were photographed using a fluorescence microscope (Nikon, Tokyo, Japan).

### 4.8. HPLC Analysis

The content of alkaloids assay was carried out by HPLC (Shimadzu, LC-20AT) on a reverse phase column (InertSustain C18, 5 µm, 4.6 mm × 250 mm) according to the method of Zhou et al. [38]. The solvent composition was: phase A, 0.03% di-n-dibutylamine in H_2_O, phase B, acetonitrile. The injection volume was 20 µL. The elution system was 0–50 min, 5–50% of B, and the column oven temperature was 35 °C with detection at 290 nm.

### 4.9. Phylogenetic Analysis

The phylogenetic tree of LaABCB11 and other ABC transporters was constructed using MEGA 7 software, following the neighbor-joining method. The Genbank access number of these proteins were listed as follow: AsABCB21 (*Apostasia shenzhenica* Z.J.Liu & L.J.Chen, PKA62294.1), DcABCB11 (*Dendrobium catenatum*, XP_020691557.1), EgABCB11 (*E. guineensis*, XP_010905015.1), CjMDR1 (*C. japonica*, BAB62040.1), AtABCG14 (*A. thaliana*, NP_564383.1), AtABCG16 (*A. thaliana*, NP_191069.2), AtABCB25 (*A. thaliana*, XP_037436171.1), AtABCC1 (*A. thaliana*, NP_001031116.1), AtABCC13 (*A. thaliana*, Q9SKX0.3), AtABCC4 (*A. thaliana*, NP_182301.1), AtABCB14 (*A. thaliana*, NP_174122.1), AtABCB1 (*A. thaliana*, NP_181228.1), AtABCB15 (*A. thaliana*, NP_189475.1), AtABCB11 (*A. thaliana*, NP_001322407.1), CjABCB2 (*C. japonica*, BAM11098.1), VvABCC1 (*Vitis vinifera* L, NP_001290005.1), NtPDR1 (*Nicotiana tabacum* L, BAB92011.1).

### 4.10. Statistical Analysis

Statistical analysis was performed using SPSS 13.0, and differences were analyzed with one-way ANOVA test or *t*-test. Statistical significance was assumed at *p* < 0.05.

## Figures and Tables

**Figure 1 ijms-22-11458-f001:**
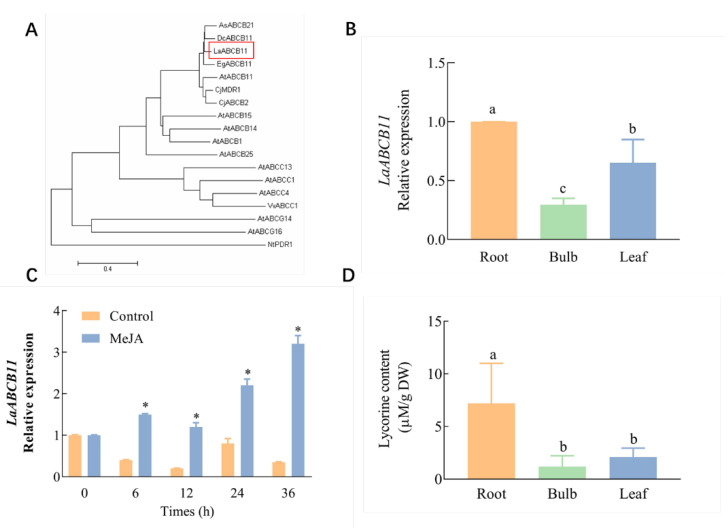
Characterization of LaABCB11 and lycorine content in different tissues of *L. aurea*. (**A**) Phylogenetic tree constructed using the MEGA 7.0 software. Sequences were found on the NCBI (https://www.ncbi.nlm.nih.gov/, 9 January 2021), LaABCB11 is boxed. (**B**) *LaABCB11* expression level in different tissues of *L. aurea*; and (**C**) under 100 µM MeJA treatment in leaves. (**D**) Lycorine content in different tissues of *L. aruea* seedling. The error bars represent standard deviations from three biological replicates, and different letters above the bars indicates significant difference determined by one-way ANOVA test (*p* < 0.05), * indicates a significant difference determined by *t*-test (*p* < 0.05), comparing with the control.

**Figure 2 ijms-22-11458-f002:**
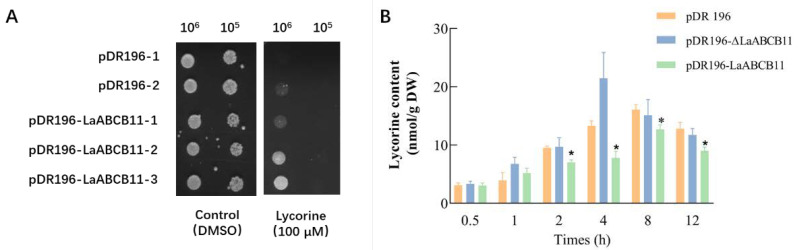
Lycorine sensitivity and export assays in yeast cells transformed with LaABCB11. (**A**) yeast mutant AD12345678 transformed with pDR196 (control) or pDR196-LaABCB11 were pre-cultured and diluted to OD_600_ = 0.1 and 0.01. Five µL of culture was dropped onto solid SD medium containing 100 µM lycorine and incubated at 28 °C for 2 days. (**B**) Yeast cells expressing pDR196 (control), pDR196-LaABCB11 or pDR196-ΔLaABCB11 (mutant of LaABCB11), were incubated in SD medium with 0.3 mM lycorine for 0.5–12 h. The error bars represent standard deviation from three biological replicates, and * indicates significant difference determined by one-way ANOVA (*p* < 0.05), compared with vector control.

**Figure 3 ijms-22-11458-f003:**
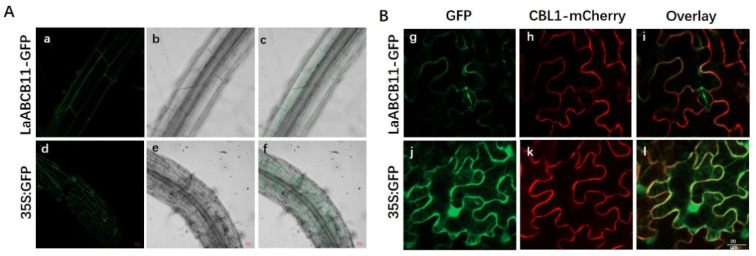
LaABCB11 localization in the cell. (**A**) Sub-cellular localization of LaABCB11-GFP in transgenic Arabidopsis root cells: (**a**) a root cell expressed LaABCB11-GFP (green); (**b**) the same cell in bright field; (**c**) merged images of (**a**,**b**); (**d**–**f**) GFP vector-expressing cells. Bars = 20 µm. (**B**) *N. benthamiana* leaf epidermal cells were transformed with the indicated plasmid combinations. Individual panels show: (**g**) an epidermal cell expressing LaABCB11-GFP (green); (**h**) the same cell expressing plasma membrane marker CBL1- mCherry (red); (**i**) merged images of (**g**,**h**); (**j**–**l**) GFP vector co-expressed with CBL1-mcherry. Bar = 20 um.

**Figure 4 ijms-22-11458-f004:**
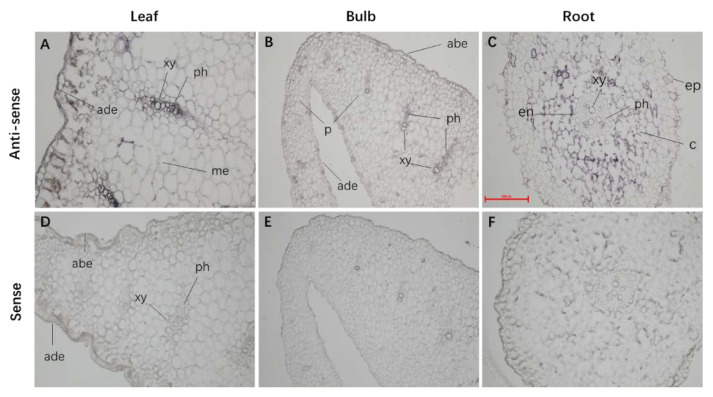
In situ hybridization of *LaABCB11* in different tissues of *L. aurea*. (**A**) Hybridizations using a digoxigenin-labeled anti-sense probe in the cross-sections of leaves, (**B**) bulbs, and (**C**) roots. (**D**–**F**), Controls: hybridization with sense probe of *LaABCB11* of tissues as indicating. xy, xylem; ph, phloem; abe, abaxial epidermis; ade, adaxial epidermis; c, cortex cell; ep, epidermis; p, parenchymal cell; en, endodermis, m, mesophyll cell. Bar = 0.25 mm.

**Figure 5 ijms-22-11458-f005:**
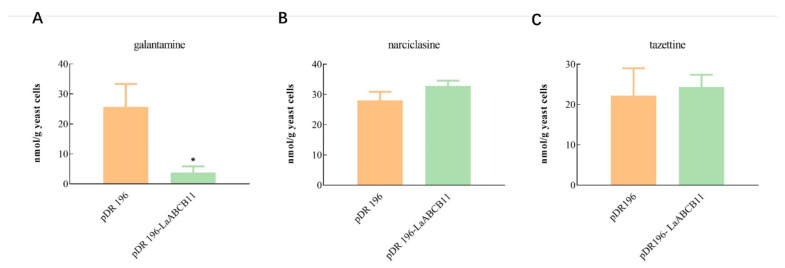
Substrates specificity for LaABCB11. (A) Yeast cell cultures expressing LaABCB11 or empty vector pDR196 were incubated with 30 µM (**A**) galantamine, (**B**) narciclasine, and (**C**) tazettine for 3 h. The amounts of substrate alkaloid accumulated in yeast cells were calculated. The error bars represent standard deviation from three biological replicates, and * indicates a significant difference determined by *t*-test (*p* < 0.05), comparing with vector control.

## Data Availability

Not applicable.

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
