# Peer review of "An ATP-Binding Cassette Transporter, LaABCB11, Contributes to Alkaloid Transport in *Lycoris aurea"

_ijms, 2021, doi:10.3390/ijms222111458_

Round 1

Reviewer 1 Report

This manuscript reports on the characterization of an enzyme that is involved in transport
of the alkaloid lycorine in the plant Lycoris aurea. The authors used a previously reported
observation that lycorine biosynthesis is upregulated by treatment with the plant hormone
methyl jasmonate as a launching point in the search for a transporter. An initial gene
gazing suggested an ABC transporter, LaABCb11, might be a suitable candidate for the
lycorine transporter. The observation that this gene was upregulated upon methyl
jasmonate treatment further supported this assumption. Hererolous expression of this
gene in yeast demonstrated that lycorine was a substrate for the protein LaABCB11. The
authors offered further support for the role of LaABCB11 by demonstrating the localization
of a GFP- LaABCB11 fusion protein in N. benthamiana. The authors then conclude this
study with some examination of the substrate specificity of LaABCB11.
Overall this is a well-executed series of experiments that characterizes the role of ABC
transporters in alkaloid biosynthesis and transport in L. aurea. There will be general
interest in this research by the community of scientists that study alkaloid biosynthesis.
The characterization of this transporter will also add some fundamental knowledge to the
field of ATP-dependant transporters.
This manuscript should be published once the very few minor changes listed below are
addressed. 

Specific Comments:
Page 2 Line 92: The phrase “phylogenetical tree” should be replaced with “phylogenetic
tree”
Gene and Protein names: The authors should ensure that they are consistent in using
lower case italizced names for genes and upper case plain font for the protein names
N. benthamiana name: The authors should provide the full taxonomic name of N.
benthamiana in the body of the text of the manuscript in the appropriate sections (Page
4 line 140 – 157)

Author Response

Firstly, we would like to thank you for your constructive comments on our article. According to your comments, we have revised the manuscript extensively, and the detailed point-by-point responses are listed below.

Page 2 Line 92: The phrase “phylogenetical tree” should be replaced with “phylogenetic tree”

Reply: Thanks for your careful checks. We have made the correction in the revised version.

Gene and Protein names: The authors should ensure that they are consistent in using lower case italizced names for genes and upper case plain font for the protein names N. benthamiana name: The authors should provide the full taxonomic name of N. benthamiana in the body of the text of the manuscript in the appropriate sections (Page 4 line 140 – 157) 

Reply: Thank you for your reminding and careful check, we have added the full name of Nicotiana benthamiana Domin in the text. But I would explain that the upper case italizeced names for genes is accepted in plant [1]. According to your remanding, we found some gene names are not italic in submitted version, we have corrected them in the revised version.

Reference:

Adebesin, F.; Widhalm, J; et al. Emission of volatile organic compounds from petunia flowers is facilitated by an ABC transporter. 2017, 356, (6345), 1386-1388, doi: 10.1126/science.aan0826

Reviewer 2 Report

In the paper "An ATP-binding cassette transporter, LaABCB11 contributes to alkaloid transport in Lycoris aurea", the authors cloned and functionally characterized an ABC transporter LaABCB11 from Lycoris aurea Herb. This study reveals the biological characters of the ABC transporter essential to alkaloid synthesis. Therefore, it is of interest to a general readership of IJMS. However, the authors should more carefully describe the results, which need major revision. The points would need to be addressed as follows.

Abstract

Page 1, L.14

Please enter a space.  (ABC) transporterfrom  ->  (ABC) transporter from

Results

Page 3, Figure 1B and D

No explanation is given for the labels, a,b,c written over the error bars.

Page 3, Figure 1C. L.111

Why does the relative expression level of the control change over time?

Page 4, Figure 2B

Figure 2A shows that the control yeast does not grow in the presence of 100 µM lycorine. In Figure 2B, the control cells are obtained even though the culture conditions differed from the case in Figure 2A. Would you please let us show the total dry weight of the recovered cells?

Page 4, Figure 3B, L 146-7

The authors described that the GFP and mCherry signals showed an obvious overlap. However, the control GFP seemed to exhibit green fluorescence on the plasma membrane, and the GFP-fusion transporter showed the fluorescence in areas other than the plasma membrane. The referee is afraid that these results are insufficient evidence to show their expression on the plasma membrane. Therefore, Western blotting may be more appropriate to show the localization on the plasma membrane.

Round 2

Reviewer 2 Report

Now, the paper is well written and structured. Therefore, the referee recommends this paper be published.